# Linking Soil Microbial Functional Profiles to Fungal Disease Resistance in Winter Barley Under Different Fertilisation Regimes

**DOI:** 10.3390/plants14203199

**Published:** 2025-10-18

**Authors:** Mariana Petkova, Petar Chavdarov, Stefan Shilev

**Affiliations:** 1Department of Microbiology and Environmental Biotechnologies, Agricultural University—Plovdiv, 12 Mendeleev Str., 4000 Plovdiv, Bulgaria; 2Plant Genetic Resources, Institute of Plant Genetic Resources “Konstantin Malkov”, 4122 Sadovo, Bulgaria; chavdarov_petar@abv.bg

**Keywords:** winter barley, fertilisation, powdery mildew, brown rust, net blotch, soil microbiome, disease resistance, sustainable agriculture

## Abstract

Barley (*Hordeum vulgare* L.) is a major fodder crop whose productivity is often reduced by phytopathogens, especially during early growth. Understanding how soil fertility management and microbial communities influence disease outcomes is critical for developing sustainable strategies that reduce fungicide dependence and enhance crop resilience. This study evaluated the resistance of the winter barley cultivar “Zemela” to powdery mildew (*Blumeria graminis* f. sp. *hordei*), brown rust (*Puccinia hordei*), and net blotch (*Pyrenophora teres* f. *maculata*). The crop was cultivated under two soil management systems—green manure and conventional—and five fertilisation regimes: mineral, vermicompost, combined, biochar, and control. Phytopathological assessment was integrated with functional predictions of soil microbial communities. Field trials showed high resistance to powdery mildew (RI = 95%) and brown rust (RI = 82.5%), and moderate resistance to net blotch (RI = 60%). While ANOVA indicated no significant treatment effects (*p* > 0.05), PCA explained 82.3% of the variance, revealing clear clustering of microbial community functions by soil management system and highlighting the strong influence of fertilisation practices on disease-related microbial dynamics. FAPROTAX analysis suggested that organic amendments enhanced antifungal functions, whereas conventional systems were dominated by nitrogen cycling. FUNGuild identified higher saprotrophic and mycorrhizal activity under organic and combined treatments, contrasting with greater pathogen abundance in conventional plots. Overall, results demonstrate that soil fertilisation practices, together with microbial functional diversity, play a central role in disease suppression and crop resilience, supporting sustainable barley production with reduced reliance on chemical inputs.

## 1. Introduction

Barley (*Hordeum vulgare* L.) is one of the earliest domesticated cereal crop and remains of major economic importance worldwide, particularly in the brewing, feed, and food industries [1]. The important nutritional value of barley straw, as well as the specificity of its grain, define it as the best and preferred concentrated feed for animals. Of all winter cereals in Bulgaria, barley ranks second in distribution after wheat. According to data from the Ministry of Agriculture and Forestry of Bulgaria, in 2024, the areas sown with barley were 196,485 ha, and the average barley yields were 5358 kg ha^−1^ [2]. Sustainable barley production relies on timely phytosanitary monitoring to manage weeds, diseases, and pests. Its productivity is significantly affected by a number of foliar diseases, especially under favourable climatic conditions and stresses. Among the most economically relevant pathogens in temperate regions are powdery mildew (*Blumeria graminis* f. sp. *hordei*), brown rust (*Puccinia hordei*), and net blotch (*Pyrenophora teres* f. *maculata*) [3,4]. These diseases reduce photosynthetic efficiency, compromise grain filling, and result in yield losses exceeding 30% under severe epidemics [5].

The management of foliar diseases in barley traditionally relies on chemical control, but environmental concerns and the development of fungicide-resistant pathogen strains necessitate alternative strategies. Breeding for disease-resistant cultivars remains one of the most effective and sustainable approaches [6]. Two main types of resistance are recognised: race-specific (vertical) resistance, which is often overcome by new pathogen races, and partial (horizontal) resistance, which is more durable and polygenic in nature [7,8]. Partial resistance is characterised by slower disease progression, reduced sporulation, and delayed symptom onset, and it plays a central role in integrated disease management and breeding programmes aimed at long-term sustainability [9]. Evaluating cultivar performance under field conditions, especially in agroecological systems, is critical for identifying genotypes with reliable resistance traits.

Soil microbial communities are increasingly recognised as key determinants of plant health, not only by contributing to nutrient cycling and soil fertility, but also by mediating disease suppression through functional diversity and ecological interactions. The soil microbiome influences host resistance by providing antagonistic activity against pathogens, competing for resources, and priming systemic plant defences [10,11]. Advances in amplicon sequencing and functional prediction tools, such as FAPROTAX and FUNGuild, have enabled the linkage of microbial community structure to ecological functions, providing new opportunities to predict crop susceptibility or resilience to foliar diseases [12,13]. Compared to traditional taxonomic profiling, these tools allow rapid, high-throughput inference of ecological functions directly from amplicon data, offering cost-effective insights into nutrient cycling, pathogenicity, and symbiotic interactions without requiring full metagenomic sequencing or culture-based assays.

In barley, host resistance and chemical control remain central strategies, and sustainable disease management increasingly emphasises the role of the rhizosphere microbiome. Agroecological practices, such as green manuring, organic amendments, and integrated fertilisation, reshape soil microbial networks, enriching functions such as chitinolysis, cellulolysis, and secondary metabolite production that actively participate in restricting pathogen development [14,15]. By contrast, conventional systems often favour microbial groups involved in nitrogen cycling, which, although important for plant nutrition, may not provide effective protection against foliar pathogens [16].

Recent evidence suggests that soil functional profiles can serve as predictors of aboveground disease outcomes. For example, enrichment of chitinolytic bacteria (*Bacillus*, *Streptomyces*) and saprotrophic fungi (*Trichoderma*, *Mortierella*) has been associated with reduced severity of powdery mildew and rust in barley, whereas conventional management correlates with a higher abundance of plant pathogenic fungi such as *Fusarium* [10,11]. Functional prediction studies confirm that agroecological soils support multifunctional microbial communities that enhance decomposition, nutrient turnover, and natural disease suppression [17,18]. Combining functional predictions of the soil microbiome with phytopathological evaluations offers an effective approach to anticipate disease dynamics in cropping systems. By connecting soil ecological processes with plant resistance, this strategy supports the development of sustainable practices that depend on chemical inputs while enhancing crop resilience.

This investigation addressed a knowledge gap regarding the role of fertilisation type in shaping disease resistance in winter barley. By integrating phytopathological assessments with microbial functional predictions, our study provides insights into how fertilisation strategies could influence the incidence and severity of foliar diseases. These findings have direct implications for the barley cultivation industry by identifying fertilisation practices that enhance crop resilience, reduce dependence on chemical fungicides, and contribute to sustainable production systems.

## 2. Results

### 2.1. Morphological Assessment

To complement quantitative assessments, foliar symptoms were evaluated morphologically to identify features associated with partial resistance (Appendix A). For *Blumeria graminis* f. sp. *hordei*, sparse mycelium, limited conidiation, and the absence of chlorosis were exhibited, indicating reduced pathogen development. Lesions caused by *Pyrenophora teres* f. *maculata* were typically small, elliptical, and bordered by chlorotic margins, with limited lesion coalescence and delayed progression, features consistent with quantitative resistance. In *Puccinia hordei*-infected plants, uredinia were small (<2 mm) and dispersed, frequently surrounded by necrotic halos, suggesting delayed sporulation and restricted pathogen spread. The morphological traits observed across all three pathogens in the present study are consistent with the expression of partial resistance and strongly support the field-based resistance index data. Such characteristics are of high agronomic value for breeding programmes aimed at achieving durable and environmentally sustainable disease resistance.

### 2.2. Disease Severity Across Barley Variants

The comparative analysis of disease severity and resistance indices (Table 1) revealed clear interactions between fertilisation strategy, disease pressure, and resistance expression in the barley cultivar “Zemela”. In the plots subjected to green manuring with a vetch–oat mixture from the previous year, most treatments demonstrated high resistance (RI ≥ 80%) to *Blumeria graminis*, *Puccinia hordei*, and *Pyrenophora teres*. Disease severity for net blotch was lowest under biochar application (11%), while powdery mildew and brown rust were effectively suppressed by mineral fertiliser, vermicompost, and combined amendments. Although moderate resistance (RI = 60–79%) was observed in some treatment–pathogen combinations, the overall pattern indicated that organic and integrated fertilisation enhanced the expression of partial resistance.

In contrast, the conventionally managed plots showed consistently higher disease severity, with multiple instances of susceptibility (RI < 40%) for powdery mildew and brown rust. Nevertheless, some treatments still provided effective suppression, such as mineral fertilisation against brown rust (95% RI) and vermicompost against powdery mildew (90% RI), suggesting that fertilisation inputs can partially compensate for less favourable soil conditions. Net blotch severity in control plots remained moderate in both systems, indicating pathogen-specific interactions with host resistance that were less dependent on fertilisation. The results demonstrate that the expression of partial resistance in cultivar “Zemela” was strongly influenced by prior soil management. Differences in disease severity reflected the influence of fertilisation type on host responses, with variability observed between organic and conventional management systems.

Green manuring supported consistently higher resistance expression, most likely due to enhanced microbial diversity, improved nutrient availability, and more favourable soil structure. Similar effects of microbiome-driven disease suppression have been reported in barley, where rhizosphere diversity reduced susceptibility to powdery mildew [19]. More broadly, organic amendments and crop diversification are known to restructure soil microbial networks and improve plant health Yang et al., 2022 [13]. Organic and integrated amendments—particularly biochar, compost plus mineral fertiliser, and vermicompost—showed superior disease suppression compared to mineral fertilisation alone. These findings underline the role of crop rotation with leguminous–cereal mixtures and biologically active soil amendments as key components of sustainable disease management in barley.

### 2.3. Effects of Cropping System and Fertilisation on Foliar Disease Severity

A one-way analysis of variance (ANOVA) was conducted to evaluate the effects of different soil management practices on the severity of three economically significant foliar diseases of barley: powdery mildew, brown rust, and net blotch. The ANOVA analysis assessing the impact of agronomic treatments on disease severity in barley demonstrated statistically significant differences (*p* < 0.05) across the tested fertilisation regimes and cropping systems for all three foliar diseases. A one-way ANOVA was conducted to evaluate the effect of fertilisation treatments on disease severity (% attack) within each cropping system (Table 2). The results were as follows: for Plot 1 (agroecological system with green manure), statistic was F = 0.139, *p*-value = 0.964. Plot 2 (conventional system without pre-cultivation) has been calculated with F-statistic = 0.205, *p*-value = 0.930. These high *p*-values (>0.05) indicate that there were no statistically significant differences in overall disease severity among the fertilisation treatments within either plot type when data for all three foliar diseases were combined.

The five fertilisation regimes produced varying levels of disease suppression (Figure 1). The unfertilized control in the conventional field (Plot 2.1) exhibited the highest disease severity, reaffirming its susceptibility due to the absence of nutritional support. In contrast, all fertilised treatments reduced disease levels to varying extents, with some organic and integrated amendments showing superior performance. Among the treatments, vermicompost (VC) and vermicompost combined with mineral fertiliser (MF.VC) had particularly strong effects in the conventional field (Plot 2), while in the agroecological field (Plot 1 with green manure), disease levels were already lower across treatments. These results suggest that the addition of organic matter and biologically active inputs enhances soil health and plant immunity, thus mitigating disease development. Biochar (BCh) also showed a statistically significant suppressive effect, especially on net blotch, which is likely attributable to improvements in soil physical and microbial properties. The cropping system itself had a notable effect. Disease severity was generally lower in Plot 1, which incorporated green manuring with oat and vetch. This practice likely enriched soil microbial diversity and improved plant vigour, leading to increased resistance against foliar pathogens. In contrast, the conventionally managed Plot 2 (without pre-cultivation) exhibited higher disease pressure under the same fertilisation regimes, underscoring the additional benefit of agroecological management. The ANOVA results confirm that both fertilisation type and field management system significantly influence the severity of key foliar diseases in barley. Organic and integrated fertilisation strategies, particularly when combined with green manuring, offer a sustainable and effective approach to disease mitigation, supporting the crop resilience in both agronomic and ecological terms.

### 2.4. Principal Component Analysis (PCA) of Disease Pressure and Soil Management

A Principal Component Analysis (PCA) in Figure 2 was conducted to explore the relationship between fertilisation treatments, cropping systems, and the severity of three major foliar diseases in barley: powdery mildew, brown rust, and net blotch. The PCA was based on standardised disease severity data and included two main components.

The first two principal components accounted for a cumulative 82.30% of the total variance in the dataset, with PC1 accounting for 48.87% and PC2 for 33.42%. This indicates that the dimensionality of the disease pressure data can be effectively reduced to two components while retaining most of the variability across treatments. The PCA revealed distinct clustering patterns based on both fertilisation and green manure application. Treatments applied in intercropped fields (Plot 1, following green manuring with intercropped oat and vetch (*Avena sativa* L. and *Vicia sativa* L.)) tended to separate from those in the conventional system (Plot 2, after fallow) along both principal components, suggesting that pre-cultivation practices significantly influenced disease pressure profiles. Among treatments, vermicompost (VC) and compost + mineral fertiliser (C.MF) showed clear separation between plots, reflecting the impact of cropping system context on treatment efficacy. Biochar (BC) and mineral fertiliser (MF) treatments clustered more closely in Plot 1 but diverged in Plot 2, indicating potential interactions between fertilisation input and soil condition or microbial environment. The positioning of the unfertilized control samples also highlighted higher variability and more pronounced disease pressure in the absence of amendments, particularly under conventional management. The PCA results underscore the importance of integrated soil fertility management and agroecological practices in shaping the fungal disease landscape in barley. The strong influence of cropping system and the clear separation of treatments in multivariate space suggest that disease outcomes are not solely dependent on the type of input but also on the biological and structural conditions of the soil ecosystem. These insights support the adoption of context-aware fertilisation strategies that leverage both organic amendments and sustainable field practices to enhance plant health and reduce disease incidence.

#### 2.4.1. Bacterial Community Shifts Across Treatments

The bacterial community composition varied between treatments (Table 3). *Lysobacter dokdonensis* was the most abundant species overall, reaching 1.32% in MF1.VC1, 1.01% in VC1, and 0.91% in Bch2. *Gemmatimonadetes bacterium* and *Gemmatimonadetes bacterium* were also consistently represented, with maxima of 0.72% and 0.56% in Bch2 and MF2.VC2, respectively. In contrast, *Jahnella thaxteri* was present at low levels across most treatments but increased markedly in MF1.VC1 (0.55%). Control 1 was enriched in *Lysobacter* (0.9%), while Control 2 contained relatively higher proportions of *Nitrospira japonica* (0.33%) and *Agromyces ramosus* (0.26%). The MF treatments influenced different taxa; MF1 favoured *Arenimicrobium luteum* (0.82%), while MF2 supported the growth of *Agromyces ramosus* (0.30%) and *Nitrospira japonica* (0.28%). Vermicompost also altered the community, VC1 increased *Lysobacter* (1.01%), and VC2 promoted *Gemmatimonadetes* species (0.45–0.40%). Combined treatments generated unique responses. MF1.VC1 strongly increased *Jahnella thaxteri* (0.55%), while MF2.VC2 enhanced both *Gemmatimonadetes bacterium* LX87 (0.66%) and LP81 (0.56%). Biochar also influenced the microbial profile: Bch1 enriched *Agromyces ramosus* (0.50%), and Bch2 showed the highest abundance of *Gemmatimonadetes* LX87 (0.72%) and *Steroidobacter* sp. (0.39%). The oat-vetch intercropping selectively enhanced taxa linked to soil health (*Lysobacter*, *Steroidobacter*) and nutrient cycling (*Nitrospira*), indicating that management practices strongly influence rhizosphere microbial structure.

#### 2.4.2. Fungal Community Shifts Across Treatments

Fungal community composition varied substantially among treatments (Table 4). *Acrophialophora jodhpurensis* reached its highest relative abundance in VC1 (24.52%), while remaining below 1% in Control 2, MF2, VC2, and MF2.VC2. *Fusarium equiseti* was consistently detected across treatments, with the greatest proportion in Bch1 (14.63%). *Humicola nigrescens* was dominant in Control 2 (11.08%), MF2 (10.10%), and MF2.VC2 (13.95%), whereas *Humicola fuscoatra* peaked in Bch2 (5.08%). *Chrysosporium lobatum* occurred across all treatments, with abundances between 1.9% and 11.0%. *Stachybotrys chartarum* was particularly enriched in Bch2 (7.22%) and MF2.VC2 (6.30%). Minor taxa included *Fungi* sp. (1.4–3.9%) and *Penicillium polonicum*, which was rare in most treatments (<0.3%) but increased in MF2.VC2 (2.72%) and Bch2 (3.29%). Organic amendments such as vermicompost and biochar strongly shaped fungal assemblages, favouring specific genera with known ecological functions, including saprotrophs (*Humicola*, *Chrysosporium*), potential pathogens (*Fusarium*, *Stachybotrys*), and beneficial antagonists (*Penicillium*).

### 2.5. Functional Annotation of Soil Microbial Communities

#### 2.5.1. Main Functions Across Soil Microbial Communities

FAPROTAX functional prediction revealed distinct differences in microbial functional profiles between the green manure field (Plot 1) and the conventional field (Plot 2) (Figure 3). In both fields, chemoheterotrophy dominated across all treatments, accounting for more than 80% of the functional potential. These functions reflect the overall prevalence of microorganisms involved in organic carbon turnover under aerobic conditions, which represent a baseline activity of soil heterotrophic microbiota. Consistent differences were observed between the two cropping systems. The field after green manure application (Plot 1) showed higher relative abundances of functions associated with chitinolysis and fermentation, particularly under vermicompost (VC1) and the combined mineral–vermicompost treatment (MF.VC1). These functions are often linked to microbial antagonism against fungal pathogens, reflecting the enrichment of chitin-degrading bacteria (e.g., *Bacillus*, *Streptomyces*) and fermentative taxa capable of producing secondary metabolites with antifungal activity. Biochar-amended plots (BCh1) also exhibited increased potential for aromatic compound degradation and fermentation, further supporting the role of organic matter incorporation in stimulating diverse metabolic functions.

By contrast, the conventional field (Plot 2) was characterised by higher contributions of functions related to the nitrogen cycle, including nitrate reduction, nitrification, and aerobic ammonia oxidation. These functions were particularly pronounced in mineral fertiliser (MF2) and biochar (BCh2) treatments, indicating a microbial community strongly oriented toward nitrogen transformations. While these processes are essential for plant nutrition, they are less directly associated with pathogen suppression and biocontrol capacity compared to the functional profile of the green manure system. These findings highlight that pre-cultivation with a vetch–oat mixture and the incorporation of organic amendments fostered a microbial functional structure more favourable to natural disease suppression, whereas conventional management primarily stimulated nitrogen cycling functions. This functional divergence underlines the role of organic matter enrichment in shaping the soil microbiome relevant to barley health and resilience.

FUNGuild annotation further revealed a marked influence of cropping system on fungal guild composition (Figure 4). Across all treatments, saprotrophs represented the dominant association, underscoring the central role of decomposers in both management systems. However, the relative balance between saprotrophs and pathotrophs differed significantly between the two fields. In the green manure field (Plot 1), higher proportions of plant pathogenic fungi (pathotrophs) were observed, particularly under vermicompost (VC1) and combined mineral–vermicompost (MF.VC1) treatments. These results suggest that while organic amendments in combination with green manuring enrich microbial diversity, they may also provide ecological niches favourable to certain foliar pathogens such as *Fusarium*, *Bipolaris*, and *Blumeria*.

Conversely, in the conventional field (Plot 2), saprotrophs constituted a larger fraction of the fungal community, reaching more than 45% relative abundance under several treatments (MF2, BCh2). This enrichment of saprotrophic fungi, including genera such as Mortierella and Trichoderma, highlights a greater functional orientation toward decomposition and organic matter recycling. While pathotrophs were still present, their relative abundance was consistently lower compared to the green manure system. These results demonstrate that green manure incorporation shifts fungal functional composition toward a higher proportion of plant pathogens, while conventional management favours saprotrophs. This variation indicates that agroecological practices not only consistently favour the growth of beneficial fungal communities but also emphasising the need to optimise organic inputs in a way that supports both disease suppression and soil fertility.

#### 2.5.2. Heatmaps of Functional Profiles of Bacteria and Fungi

A heatmap of bacterial functions revealed clear clustering of treatments according to soil management (Figure 5). Treatments following green manure were enriched in functions related to organic matter degradation (cellulolysis, xylanolysis), antifungal activity (chitinolysis), and secondary metabolism (fermentation, aromatic compound degradation). The chitinolysis was especially promoted in combined treatment together with chemoheterotrophy, especially aerobic. In addition, the organic treatment (VC) expressed most abundant functions, such as those typically attributed to the inorganic fertilised treatments: nitrate and iron respiration; nitrite and nitrate ammonification driven by nitrifying and denitrifying taxa such as *Nitrosospira multiformis*, *Nitrobacter winogradskyi*, *Paracoccus denitrificans*, and *Pseudomonas stutzeri* [16]. While these functions improve nutrient turnover, they are less directly linked to pathogen suppression. Mineral fertilisation treatment was relatively poor of functional groups. Biochar treatment in this field showed higher abundance of functions related to the nitrogen cycling, such as aromatic compound degradation, nitrification, and ammonia oxidation and nitrate reduction. In contrast, the functional groups of second field (after fallow) were more dependent of amendment than the other one. Thus, in the control, nitrification, nitrate reduction, ureolysis, and chloroplasts were the most abundant functions showing soil ecosystem functioning without external nutritional inputs. In MF, abundant photoheterotrophic and photoautotrophic functions were found, probably because of the need of carbon acquisition and luck of sufficient organic matter. When VC was added, some functions of the nitrogen cycling were affected. The functions related to the photosynthesis and N respiration were significantly expressed in the treatment with biochar.

The fungal functional heatmap (Figure 6) revealed clear differences between management systems. In the green manure field (Plot 1), vermicompost (VC1) and the combined mineral–vermicompost (MF.VC1) showed higher relative abundances of plant pathogenic microbes and mixed functional groups combining pathogenic and saprotrophic roles. These results indicated that organic amendments under green manure conditions may stimulate both beneficial and detrimental fungal taxa.

By contrast, the conventional field (Plot 2) was characterised by greater enrichment of wood, soil and leaf saprotrophic fungi. These functional groups were particularly pronounced in the MF2 and BCh2 treatments, consistent with the higher relative dominance of saprotrophs observed in Figure 4. The overall reduction in pathogenic microbes in conventional plots suggests a more decomposition-oriented fungal community compared to the pathogen-enriched structure under green manure. These functional differences between management systems demonstrate that while green manure incorporation promotes multifunctional fungal groups—including antagonists, symbionts, and pathogens—conventional management supports communities more strongly oriented toward organic matter recycling. These findings emphasise the need to balance organic amendments in agroecological systems to enhance beneficial fungal functions while constraining the proliferation of plant pathogens.

#### 2.5.3. PCA of Bacterial Functional Annotation Profiles

PCA of the functional profiles predicted by FAPROTAX revealed clear separation of treatments along the first two principal components (PC1 = 31.9%, PC2 = 17.1%) (Figure 7). Green fertiliser treatments (Control1, MF1, VC1, MF1.VC1) tended to cluster separately in PC1, except Bch1. On the other hand, the conventional treatments (Control2, MF2, VC2, MF2.VC2, Bch2) clustered a part, indicating that soil management changed the microbial functional structure. BCh1 and BCh2 were very close situated concerning the PC1 suggesting important influence of its physicochemical properties on bacteriome structure and functions.

PCA based on FUNGuild assignments revealed a clear separation of treatments along PC1 (43.83%) and PC2 (19.71%) (Figure 8). Some treatments of field 1 (VC1, MF1.VC1, Bch1) were clustered apart from conventional treatments (Control2, MF2, VC2, MF2.VC2), indicating distinct fungal functional profiles shaped by management. In particular, VC1 and Bch1 were positioned at the extremes of PC1 and PC2, reflecting strong enrichment of saprotrophic and mycorrhizal fungi under organic–integrated fertilisation and biocontrol amendments. Conversely, conventional treatments grouped more closely, suggesting functional homogenisation dominated by plant pathogenic and undefined species.

## 3. Discussion

The morphological characterisation of disease symptoms substantiated the expression of partial resistance in the barley cultivar “Zemela”. Restricted mycelial development and limited conidiation in *Blumeria graminis* f. sp. *hordei*, slowly expanding lesions in *Pyrenophora teres f. maculata*, and small uredinia surrounded by necrotic halos in *Puccinia hordei* were consistent with traits of resistance [7,9]. Such resistance is less prone to breakdown and provides greater stability under variable environmental conditions, particularly when combined with soil health-promoting practices [8,20,21,22].

This study demonstrates that the “Zemela” cultivar of barley expresses stable field resistance against three major foliar pathogens—*B. graminis* f. sp. *hordei*, *Puccinia hordei*, and *Pyrenophora teres* f. *maculata* (Table 1). Disease outcomes varied markedly across fertilisation regimes and cropping system histories, underscoring the role of agronomic context and nutrient input in shaping resistance expression. In green manure plots, particularly those amended with biochar and mineral fertiliser, resistance indices (RI) frequently exceeded 80% for all pathogens. By contrast, vermicompost and combined compost and mineral fertiliser achieved only moderate resistance levels. These patterns are consistent with the concept that intercropping and organic inputs enhance plant health through improved nutrient availability, increased rhizosphere microbial diversity, and activation of systemic resistance [7,9].

In conventional plots lacking prior green manure, disease severity was substantially higher (Table 1). Susceptibility was most pronounced to powdery mildew and brown rust, with RI values often dropping below 40%. Certain fertilisation strategies, such as mineral fertiliser (Plot 2.2) for brown rust and vermicompost (Plot 2.3) for powdery mildew, maintained relatively high resistance, highlighting the influence of nutrient form and soil microbial context on host defence [21,22,23]. Comparisons across fertilisation strategies confirmed consistent trends in the expression of disease resistance. Under green manuring, the unfertilized control (Plot 1.1) showed moderate resistance to net blotch (RI = 67%) and high resistance to powdery mildew (90%) and brown rust (87%). By contrast, the conventional control (Plot 2.1) displayed strong susceptibility, particularly to powdery mildew and brown rust (RI = 45% and 20%, respectively). These results emphasise the legacy effects of green manure, which enhance microbially mediated suppression of biotrophic pathogens even under nutrient-limited conditions [24].

Mineral fertiliser (Plots 1.2, 2.2) consistently conferred high resistance (RI 80–95%) across all diseases, reflecting the importance of balanced NPK nutrition in maintaining basal defences (Table 1). Brown rust resistance peaked at 95% under conventional conditions (Plot 2.2), suggesting that improved leaf tissue integrity and metabolic status contribute to enhanced resistance [23,24].

The performance of vermicompost (Plots 1.3, 2.3) was more variable. In the green manure field, it maintained high resistance to powdery mildew (65–88%) and brown rust (80%), but only moderate resistance to net blotch (75%). In the conventional field, resistance declined sharply, with high susceptibility to net blotch and brown rust (RI = 30% and 24%). This indicates that vermicompost is more effective in biologically enriched soils, where synergistic interactions with diverse microbial communities enhance induced systemic resistance [21].

The combined compost–mineral fertiliser treatment (Plots 1.4, 2.4) provided consistent and balanced resistance across systems, though slightly stronger under green manure conditions (Table 1). Powdery mildew resistance, for example, reached 82% in Plot 1.4 compared to 70% in Plot 2.4. This supports earlier findings that integrated nutrient strategies deliver dual benefits: sustained nutrition and stimulation of rhizosphere immunity [9,22].

Biochar amendments revealed contrasting outcomes. In Plot 1.5 (green manure), resistance ranged from 68 to 89% across all pathogens, while in Plot 2.5 (conventional), resistance remained high to brown rust (RI = 90%) and moderate to net blotch (75%), but dropped markedly for powdery mildew (RI = 28%). The porous structure of biochar improves soil aeration and microbial activity, supporting suppression of necrotrophic pathogens such as *Pyrenophora teres* under favourable soil conditions [21,22].

Across both fields, treatment effects were more consistent and informative than plot-level differences, confirming the importance of fertilisation type as the primary driver of plant–pathogen–microbiome interactions. Vermicompost and combined fertilisation enhanced partial resistance against powdery mildew, brown rust, and net blotch, and supported enrichment of microbial taxa with known antifungal potential, such as *Penicillium* and *Lysobacter*. Biochar also contributed to disease suppression, most likely through indirect effects on soil structure, nutrient retention, and the creation of microsites favouring beneficial microbes. In contrast, mineral fertilisation improved crop nutrition and promoted nitrifying bacteria such as *Nitrospira*, but its effects on pathogen resistance were less pronounced.

PCA confirmed a clear divergence in microbial functional profiles between management systems: green manure promoted functions linked to disease suppression, while conventional management favoured nitrogen transformation pathways (Figure 2). These results are consistent with the observed reductions in powdery mildew and brown rust severity in agroecological plots, supporting the functional role of microbiomes in mediating crop resilience [10,11,14].

The results presented in Table 3 demonstrate that soil amendments substantially influenced the relative abundance of key bacterial taxa. The consistent enrichment of *Lysobacter dokdonensis* in VC1, MF1.VC1, and Bch2 suggest a strong response to organic and combined inputs. Members of the genus *Lysobacter* are known for their production of extracellular enzymes and antimicrobial metabolites, which contribute to disease suppression in soils [24,25]. Similarly, the higher representation of *Steroidobacter* sp. in biochar treatments may indicate enhanced degradation of complex organic compounds, a function frequently reported for this genus [26,27]. The occurrence of *Nitrospira japonica* at elevated levels in Control 2 and MF2 highlights the importance of nitrifying bacteria in untreated and fertilised soils. *Nitrospira* plays a central role in nitrite oxidation and has been associated with improved nitrogen turnover under both conventional and amended systems [28]. The selective increase of Jahnella *thaxteri* in MF1.VC1 is noteworthy, as this taxon was nearly absent in other treatments, suggesting that microbial consortia can create narrow ecological niches favouring less common bacteria. The observed shifts underline that microbial fertilisers, vermicompost, and biochar act not only as nutrient sources but also as ecological drivers that shape bacterial assemblages. By promoting taxa with known biocontrol and nutrient-cycling potential, these amendments may contribute to improved soil functionality and crop resilience.

The fungal profiles presented in Table 4 highlight the strong influence of soil amendments on community composition. *Acrophialophora jodhpurensis* was highly enriched in VC1 (24.52%), suggesting that vermicompost favoured the proliferation of this saprotrophic genus, which has been reported to degrade complex substrates and contribute to organic matter turnover [29]. The dominance of *Fusarium equiseti* in several treatments, particularly Bch1 (14.63%), is of concern, as this species is associated with plant pathogenicity and mycotoxin production [30]. Amendments also promoted species of *Humicola*, with *H. nigrescens* showing high relative abundance in Control 2, MF2, and MF2.VC2. *Humicola* species are thermophilic decomposers involved in cellulose degradation, often enriched under organic inputs [31]. Biochar amendments notably increased Stachybotrys chartarum (7.22% in Bch2), a known opportunistic fungus, but also enhanced *Humicola fuscoatra* (5.08%), indicating that biochar may create favourable microsites for both beneficial and potentially harmful taxa. *Penicillium polonicum* occurred at low levels across most treatments, its enrichment in MF2.VC2 (2.72%) and Bch2 (3.29%) are noteworthy, as *Penicillium spp.* are recognised for their antagonistic activity against soilborne pathogens [32]. Current results demonstrate that fertilisation practices significantly altered rhizosphere microbial communities, which in turn correlated with foliar disease resistance. For example, vermicompost and combined amendments promoted enrichment of *Lysobacter* and *Penicillium* spp., taxa with known antifungal potential, aligning with lower severity of powdery mildew. Conversely, mineral fertilisation increased nitrifying bacteria such as *Nitrospira*, which supported nutrient turnover but was less directly associated with pathogen suppression. These findings underline the importance of fertiliser-driven shifts in microbial ecology as a mechanistic link to plant disease resistance.

Our results demonstrate that soil fertilisation practices influence not only foliar disease outcomes but also the composition and predicted functions of rhizosphere microbial communities in winter barley. Although ANOVA indicated no statistically significant effects, multivariate analyses (PCA) and functional prediction tools (FAPROTAX, FUNGuild) revealed clear ecological trends, suggesting that fertilisation type and preceding crop management shape disease dynamics through shifts in microbial community functions. These findings highlight the importance of considering both statistical and biological significance when interpreting complex agroecosystem interactions. Functional predictions from FAPROTAX and FUNGuild provide valuable, hypothesis-generating insights into microbial roles in nutrient cycling and pathogen suppression. However, these predictions are not confirmatory, and the observed associations should be interpreted as potential rather than causal mechanisms. Further metagenomic, transcriptomic, or culture-based studies are needed to validate the contribution of specific microbial groups to disease resistance in barley.

In this study, the functional predictions suggested possible mechanistic explanations for the observed patterns. FAPROTAX indicated that green manure tended to enrich functions related to organic matter turnover, chitinolysis, and aromatic compound degradation, largely associated with taxa such as *Bacillus* and *Streptomyces*. These functions are often linked in the literature to antifungal enzyme production and biocontrol potential [33]. In contrast, conventional management was associated more strongly with nitrogen cycling functions, including nitrification and nitrate reduction, which, while beneficial for plant nutrition, are less directly connected to pathogen suppression. Similarly, FUNGuild analyses suggested that green manure plots harboured a higher proportion of pathotrophic fungi, whereas saprotrophic guilds were relatively more abundant under conventional management. Vermicompost under green manure particularly increased the relative abundance of pathotrophs, while conventional treatments such as MF2 and BCh2 appeared to favour saprotrophs, including *Mortierella* and *Trichoderma*. Heatmap analyses further reflected these differences, highlighting a potential shift toward multifunctional but pathogen-enriched fungal communities under green manure compared with the more saprotroph-dominated profiles of conventional systems.

Escudero-Martinez et al. (2022) [34] showed that specific barley genes shape rhizosphere microbiota composition, underscoring the importance of plant genotype in steering microbial communities. Together, these studies reinforce the hypothesis-generating functional predictions presented here and underscore the potential of soil and crop management practices, combined with host genetic factors, to steer microbial communities toward enhanced disease suppression. Recent studies in cereals support these observations. Joshi et al. (2025) [35] reviewed microbiome manipulation strategies and highlighted how bioengineering approaches, including organic amendments, can enhance plant performance and disease suppression through shifts in microbial consortia. Pokharel et al. (2025) [36] demonstrated that leguminous cover crops increased rhizosphere microbial diversity in tea, but similar mechanisms of microbiome diversification are likely relevant for cereals under sustainable management. Weerasinghe et al. (2025) [37] used metabarcoding to compare bacterial endophytes in barley grains infected and non-infected with *Fusarium head blight*, revealing distinct endophytic community shifts associated with disease status. Together, these studies reinforce the hypothesis-generating functional predictions presented here and underscore the potential of soil and crop management practices to steer microbial communities toward enhanced disease suppression.

These findings suggest that while direct statistical evidence of treatment effects was limited, biologically meaningful patterns were apparent in microbial functional shifts and disease resistance responses. Integrating predictive tools with confirmatory molecular approaches in future work will be essential for fully elucidating the mechanisms underlying barley resilience under different fertilisation strategies. Integrating field disease assessments with microbial functional predictions demonstrates that soil microbiome diversity and function play a pivotal role in stabilising crop health. The high resistance index values for “Zemela” cult. of barley (95% against powdery mildew, 82.5% against brown rust, 60% against net blotch) align with the enrichment of chitinolytic and antifungal microbial taxa in green manure plots. These findings highlight Zemela’s suitability for sustainable systems, where the synergy between plant genotype and soil microbiota underpins durable disease resistance. Agroecological practices such as green manure, organic amendments, and integrated fertilisation not only improve soil health but also reinforce natural disease suppression, reducing reliance on chemical inputs while maintaining productivity and resilience.

Organic and integrated fertilisation likely enhances resistance by multiple pathways: (i) stimulating microbial production of antifungal metabolites and cell-wall degrading enzymes (e.g., chitinases, glucanases), (ii) improving nutrient balance and plant vigour, which strengthens basal immunity, and (iii) priming induced systemic resistance via microbial signalling molecules. In contrast, mineral fertilisation alone favoured nitrogen-cycling microbes, improving nutrition but offering limited biocontrol potential. Biochar, with its porous structure, may create microsites for beneficial microbes and sorb pathogen propagules, thereby indirectly reducing disease incidence. These mechanisms collectively explain the observed differences in resistance expression across treatments and highlight how targeted fertilisation strategies can reinforce the natural disease suppression potential of the rhizosphere microbiome.

## 4. Materials and Methods

### 4.1. Field Description and Experimental Design

The experiment was conducted in two fields, which differed only in the predecessor crop. The soils at both experimental sites were classified as Fluvisols or Chernozems with a loam texture. One field was established after an oat–vetch mixture incorporated as green manure, while the other followed a fallow period. The oat–vetch intercrop was selected as it represents a widely used cover crop mixture in sustainable farming systems, known to improve soil structure, increase nitrogen availability, and stimulate beneficial microbial activity. In contrast, the fallow field served as a baseline representing conventional practice without organic matter input. This contrast allowed us to assess how green manuring versus bare fallow influences subsequent disease pressure and soil microbial community responses in barley. Winter fodder barley (*Hordeum vulgare* L., cv. Zemela) was sown at the beginning of November 2022 at a seed density of 160 kg ha^−1^. Field 1 was established after incorporation of an oat–vetch mixture used as green manure, incorporated into the soil at the ripening phase (end of June). Field 2 was established on fallow land left without crops in the preceding season (Table 5). Each field was arranged as a randomised block design with five treatments and three replicates per treatment (15 plots per field, 30 plots in total).

The treatments were as follows: Control—no fertilisation; MF—mineral fertiliser NPK (15:15:15) at 0.1 t ha^−1^; VC—vermicompost at 12 t ha^−1^; MF.VC—combined mineral fertiliser and vermicompost at 0.05 + 6 t ha^−1^; BCh—biochar at 10 t ha^−1^. The amendments were incorporated into the upper 0–30 cm soil layer before sowing using a cultivator. Ammonium nitrate (NH_4_NO_3_, 34.4% N; Agropolychim AD, Devnya, Bulgaria) was applied at the tillering stage at a rate of 50 kg N ha^−1^ as part of the MF treatment. The alterations were produced as follows: vermicompost was prepared from cow manure and wheat straw in a ratio of 72.7%:27.3% (w:w). The process involved two stages—three months of aerobic composting, followed by vermicomposting. After the active composting phase, the resulting fresh compost was subjected to earthworm populations (Eisenia fetida and Lumbricus rubellus). Biochar derived from oat biomass was kindly provided by the Warsaw University of Life Sciences (SGGW, Poland). The physicochemical properties of vermicompost and biochar were described previously [38].

The experiment was conducted in two fields, which differ only by the predecessor crop (Table 5). Each field was organised as five treatments with four replicates and randomised block design as follows: control (no fertilisation), mineral fertiliser (N:P:K) 0.1 t ha^−1^ (MF, 15:15:15); vermicompost (VC) 12 t ha^−1^, combined mineral fertiliser and vermicompost (C.MF) 0.05 + 6 t ha^−1^, biochar (BCh) 10 t ha^−1^. The amendments were mixed into the upper soil layer (0–30 cm) before sowing through a cultivator. Ammonium nitrate (NH_4_NO_3_, 34.4% N, Agropolyhim AD, Devnya, Bulgaria) at a concentration of 50 kg/ha of nitrogen was applied at tillering phase in the mineral fertilisation treatment as a part of the fertilisation of barley.

### 4.2. Phytopathological Assessment

Morphological characteristics of disease symptoms were recorded during field observations to complement visual severity estimates and support the identification of resistance mechanisms. For each pathogen, three symptomatic leaves per plot were collected during the heading stage (Zadoks 50–59) and examined using a stereomicroscope (Leica EZ4, Leica Microsystems, Wetzlar, Germany) to observe lesion morphology, fungal structures, and tissue responses (Appendix A). Indicators of host resistance included restricted lesion expansion, necrotic flecking, and reduced sporulation. By contrast, susceptible plants were characterised by extensive lesion development, chlorosis, and profuse sporulation, with abundant reproductive structures such as conidiophores and conidia (*Blumeria graminis*), uredinia with dense urediniospores (*Puccinia hordei*), or large necrotic lesions with visible pycnidia (*Pyrenophora teres*). Disease incidence was recorded for three major foliar pathogens: powdery mildew (*Blumeria graminis* f. sp. *hordei*), net blotch (*Pyrenophora teres*), and brown rust (*Puccinia hordei*). Observations were made during the heading stage (Zadoks scale 50–59) under natural infection pressure. Three representative plants were randomly selected from each plot [39]. Disease severity was visually estimated by manual assessment using the standard James scale [40], which expresses the percentage of infected leaf area on the flag leaf and the second upper leaf. The mean disease severity (A) for each treatment was calculated asA = (1/n) ∑A_i_, where A_i_ is the individual plant score, and n is the number of observations.

The resistance index (RI) was then derived asRI = 100 − A.

Resistance was classified into four categories:RI ≥ 80%: High resistance;60% ≤ RI < 80%: Moderate resistance;40% ≤ RI < 60%: Low resistance;RI < 40%: High susceptibility.

### 4.3. Soil DNA Extraction, Amplicon Sequencing, and Bioinformatic Analysis

Soil samples were collected from ten experimental plots (five treatments under green manure management and five treatments under conventional management). Soil samples were collected as composite rhizosphere soil following the protocol described by Kirse et al. (2021) and Petkova et al. (2025) [41,42]. For each treatment, composite rhizosphere soil was obtained by pooling subsamples from five randomly selected plants at the flowering stage. Samples were transported on ice and stored at −80 °C until further processing.

#### 4.3.1. DNA Extraction

Total community DNA was isolated from 0.5 g of soil per sample using the DNeasy PowerSoil Pro Kit (Qiagen, Hilden, Germany) according to the manufacturer’s instructions. DNA concentration and purity were verified with NanoDrop spectrophotometry (Thermo Fisher Scientific, Waltham, MA, USA) and Qubit 4.0 fluorometry (Invitrogen, Carlsbad, CA, USA). Integrity was confirmed by agarose gel electrophoresis [43].

#### 4.3.2. Amplicon Library Preparation

For bacteria, the V3–V4 hypervariable regions of the 16S rRNA gene were amplified using primers 341F (5′-CCTACGGGNGGCWGCAG-3′) and 806R (5′-GACTACHVGGGTATCTAATCC-3′) [44,45]. For fungi, the ITS2 region was amplified using primers ITS3 (5′-GCATCGATGAAGAACGCAGC-3′) and ITS4 (5′-TCCTCCGCTTATTGATATGC-3′) [46]. PCR reactions were performed in triplicate for each sample in 25 µL volumes containing 12.5 µL 2× KAPA HiFi HotStart ReadyMix (Roche, Basel, Switzerland), 0.5 µM of each primer, and 10 ng template DNA. Cycling conditions included an initial denaturation at 95 °C for 3 min, followed by 25 cycles of 95 °C for 30 s, 55 °C for 30 s, and 72 °C for 30 s, with a final extension at 72 °C for 5 min. Replicate PCR products were pooled, purified with AMPure XP beads (Beckman Coulter, Brea, CA, USA), and quantified using Qubit dsDNA HS Assay Kit (Invitrogen, Carlsbad, CA, USA).

#### 4.3.3. Library Construction and Sequencing

Amplicon libraries were constructed using the Illumina Nextera XT kit and sequenced on a MiSeq platform (2 × 300 bp). Raw sequences were processed in QIIME2, with quality filtering, denoising, and chimaera removal. Amplicon sequence variants (ASVs) were assigned taxonomically against the SILVA (for the bacteria) and UNITE (for the fungi) reference databases [12,47]. Relative abundance (%) of each taxon was calculated as the proportion of reads assigned to that taxon divided by the total reads per sample, multiplied by 100. The resulting values were averaged per treatment and used to compare microbial community composition across fertilisation regimes.

Sequencing of 16S rRNA and ITS amplicons produced a total of 522,297 quality-filtered reads across 10 samples, with read counts ranging from 47,851 to 62,361 per sample (average ~52,230). These data provided sufficient depth for downstream taxonomic, diversity, and functional analyses.

### 4.4. Functional Annotation of Microbial Communities

Functional profiles of bacterial and archaeal communities were predicted using the FAPROTAX pipeline [48]. Briefly, representative Amplicon Sequence Variants (ASVs) obtained from amplicon sequencing were taxonomically assigned against the SILVA database. The resulting taxonomic profiles were then mapped to the FAPROTAX v1.2 functional database, which links prokaryotic taxa to putative metabolic and ecological functions based on curated literature. Functional groups were then clustered by abundance and visualised as bar plots, heatmaps, and principal component analysis (PCA) plots using R (v 4.3.1). Fungal ecological functions were predicted using FUNGuild based on ITS amplicon sequencing data classified against the UNITE database [12]. Relative abundances were calculated per treatment, and results were visualised with bar plots and clustered heatmaps. This allowed the identification of dominant microbial functions across treatments and comparison of functional community structure under different soil management regimes.

### 4.5. Statistical Analysis

Basic statistical analysis was applied using the tools of Microsoft Excel version 2010. In addition, statistical analysis was conducted using one-way ANOVA to identify significant differences among treatments and cropping systems. Principal component analysis (PCA) and *k*-means clustering (k = 3) were used to explore patterns in disease severity profiles. All data analyses were performed using R software (version 4.3.1, accessed on 12 July 2025), with the packages stat and ggplot2.

## 5. Conclusions

The barley cultivar “Zemela” demonstrated strong resistance to powdery mildew and brown rust, and moderate resistance to net blotch under natural infection conditions. While fertilisation treatments alone did not result in statistically significant differences in disease severity, PCA revealed clear clustering based on cropping system and soil fertilisation practices, highlighting the interactive effects of agronomic context and disease response. The green manure application and organic amendments consistently resulted in lower disease severity compared to conventional management. These results underline the importance of integrating soil fertility strategies and sustainable agronomic practices to enhance disease resistance and reduce dependence on chemical control. The combination of field evaluation, statistical analysis, and multivariate techniques provided a comprehensive assessment of cultivar performance and offers valuable insight for sustainable barley production. Future studies should incorporate molecular analyses to identify resistance genes (e.g., *mlo*, *rph*) and further characterise the genetic basis of partial resistance, supporting precision breeding and regional adaptation. Forthcoming research should integrate qPCR or metabarcoding of foliar pathogens to better link soil microbial functions with disease outcomes.

## Figures and Tables

**Figure 1 plants-14-03199-f001:**
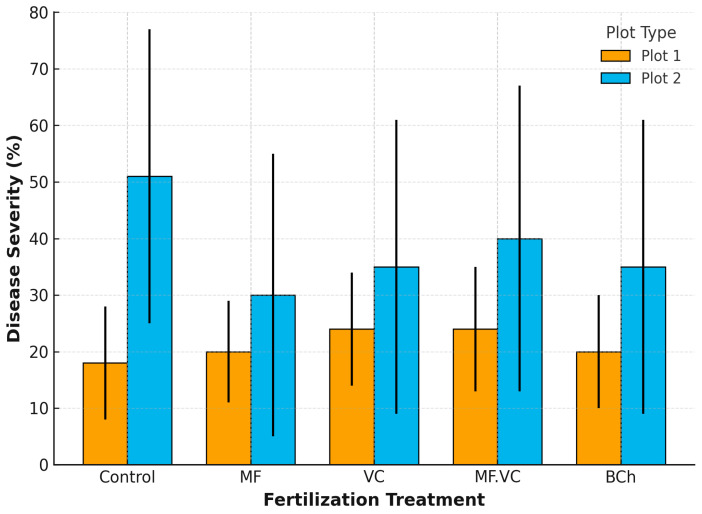
Summary of One-Way ANOVA results for disease severity across fertilisation treatments by plot type. Control—unfertilised; MF—mineral fertiliser; VC—vermicompost; MF.VC—mineral fertiliser + vermicompost; BCh—biochar.

**Figure 2 plants-14-03199-f002:**
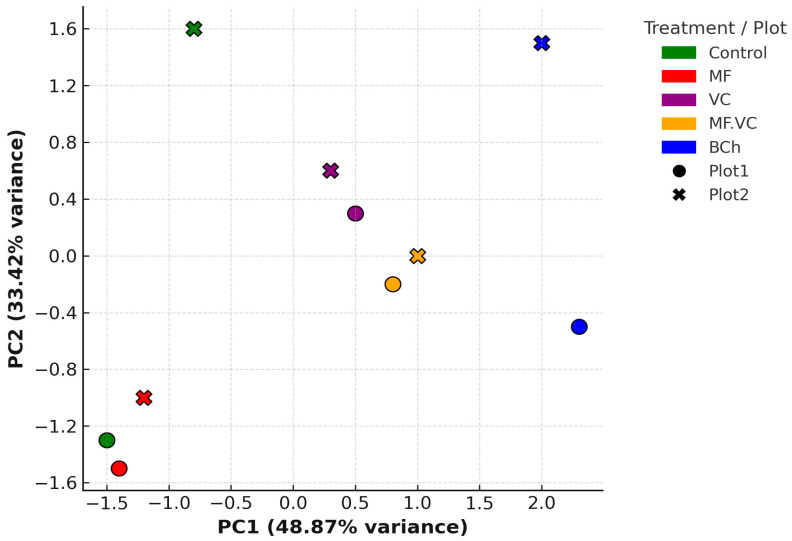
Principal coordinate analysis (PCoA) based on Bray–Curtis dissimilarity illustrating the fungal community composition under different fertilisation treatments: Control (unfertilised), MF (mineral fertiliser), VC (vermicompost), MF.VC (mineral fertiliser + vermicompost), and BCh (biochar). Each treatment was assessed in two experimental plots (Plot 1, circles; Plot 2, crosses). The first two principal coordinates (PC1 and PC2) explain 48.87% and 33.42% of the total variance, respectively. The analysis shows clear clustering patterns, indicating that fertilisation strategy influenced fungal community structure.

**Figure 3 plants-14-03199-f003:**
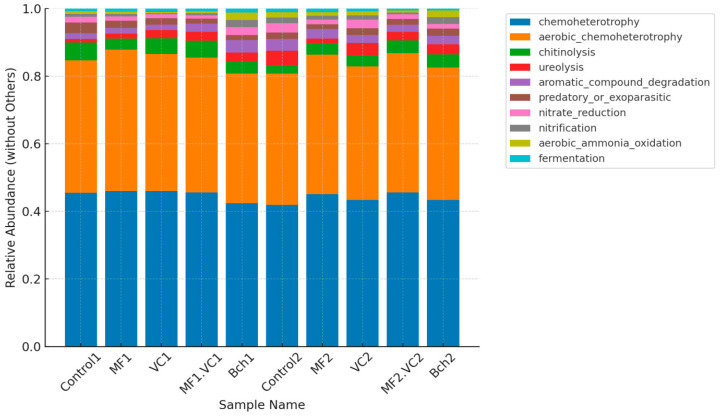
Bar plot of FAPROTAX function annotation showing main functions of soil bacteriome. Control; MF—mineral fertiliser; VC—vermicompost; MF.VC—mineral fertiliser + vermicompost; BCh—biochar. 1—refers to the field after green manure; 2—refers to the field after fallow.

**Figure 4 plants-14-03199-f004:**
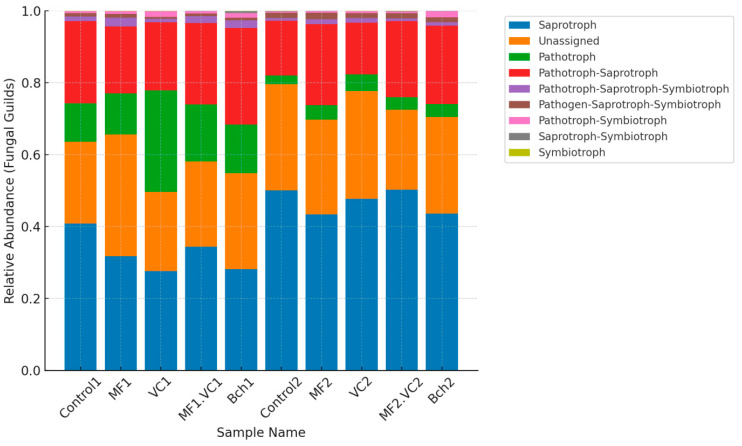
Bar plot of FUNGuild function annotation showing main functions of soil fungi. Control; MF = mineral fertiliser; VC = vermicompost; MF.VC—mineral fertiliser + vermicompost; BCh—biochar. Index 1 refers to the field after green manure, while index 2 refers to the field after fallow.

**Figure 5 plants-14-03199-f005:**
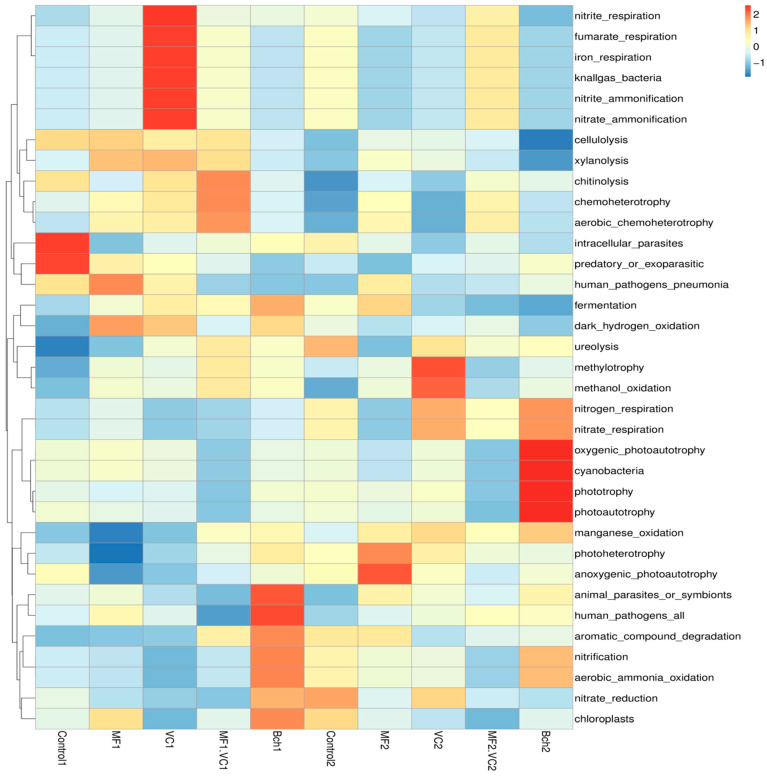
Heatmap of functional groups (FAPROTAX) across soil management treatments. Control; MF—mineral fertiliser; VC—vermicompost; MF.VC—mineral fertiliser + vermicompost; BCh—biochar. Index 1 refers to the field after green manure, while index 2 refers to the field after fallow.

**Figure 6 plants-14-03199-f006:**
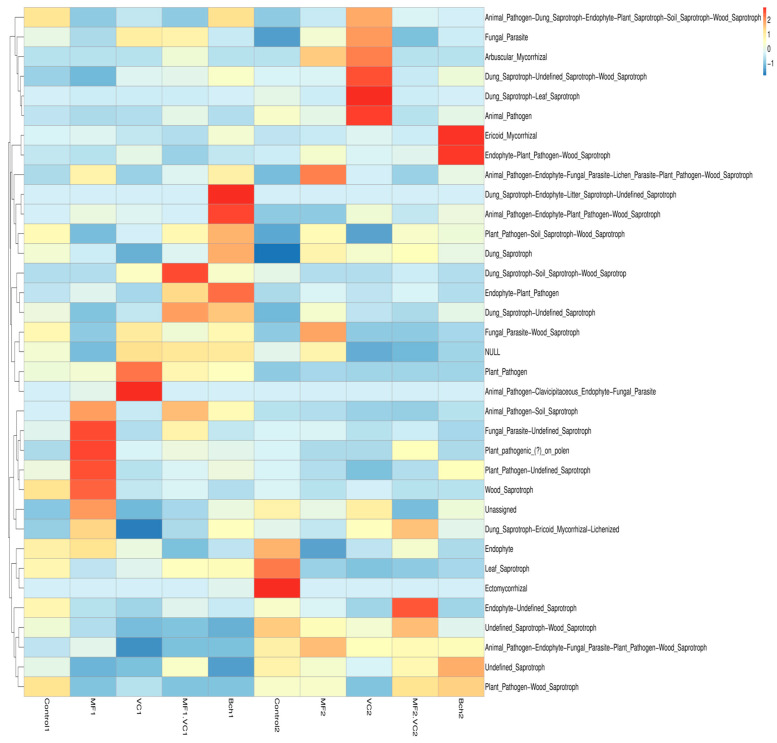
Heatmap of FUNGuild function annotation across soil management treatments. Control; MF: mineral fertiliser; VC: vermicompost; MF.VC: mineral fertiliser + vermicompost; BCh: biochar. Index 1 refers to the field after green manure, while index 2 refers to the field after fallow.

**Figure 7 plants-14-03199-f007:**
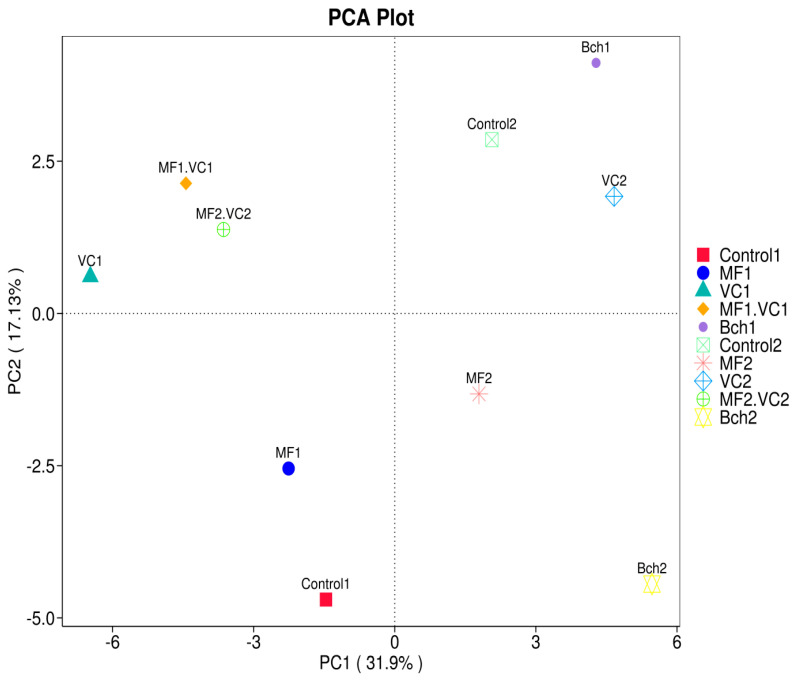
PCA result of FAPROTAX function annotation. Control; MF—mineral fertiliser; VC—vermicompost; MF.VC—mineral fertiliser + vermicompost; BCh—biochar. Index 1 refers to the field after green manure, while index 2 refers to the field after fallow.

**Figure 8 plants-14-03199-f008:**
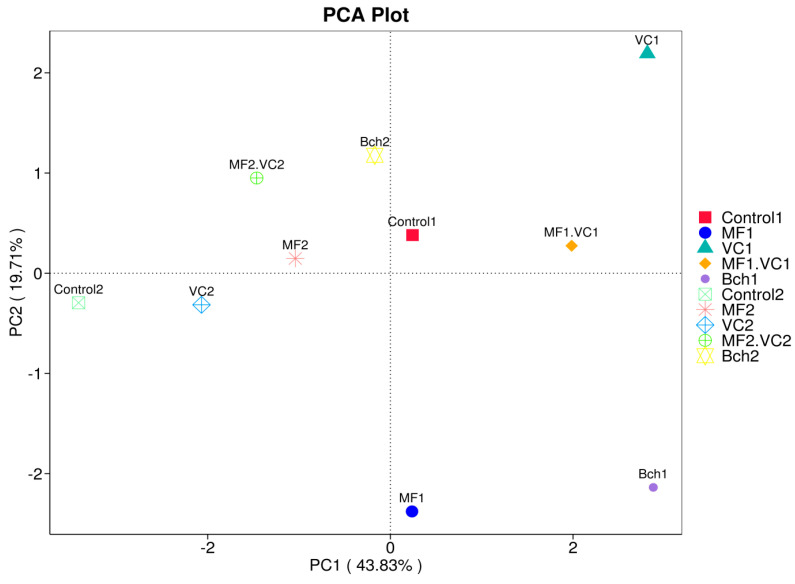
PCA result of FUNGuild function annotation. Control; MF—mineral fertiliser; VC—vermicompost; MF.VC—mineral fertiliser + vermicompost; BCh—biochar. Index 1 refers to the field after green manure, while index 2 refers to the field after fallow.

**Table 1 plants-14-03199-t001:** Disease severity (%) and resistance index (RI) for cultivar “Zemela” across treatments. Control; MF: mineral fertiliser; VC: vermicompost; MF.VC: mineral fertiliser + vermicompost; BCh: biochar.

**Plot**	**Treatment**	**Disease**	**Severity (%)**	**RI (%)**	**Resistance Category**
Plot 1.1	Control	Powdery mildew	10	90	High resistance
Plot 1.1	Control	Net blotch	33	67	Moderate resistance
Plot 1.1	Control	Brown rust	13	87	High resistance
Plot 1.2	MF	Powdery mildew	12	88	High resistance
Plot 1.2	MF	Net blotch	20	80	High resistance
Plot 1.2	MF	Brown rust	31	69	Moderate resistance
Plot 1.3	VC	Powdery mildew	35	65	Moderate resistance
Plot 1.3	VC	Net blotch	25	75	Moderate resistance
Plot 1.3	VC	Brown rust	12	88	High resistance
Plot 1.4	MF.VC	Powdery mildew	15	85	High resistance
Plot 1.4	MF.VC	Net blotch	38	62	Moderate resistance
Plot 1.4	MF.VC	Brown rust	20	80	High resistance
Plot 1.5	BCh	Powdery mildew	18	82	High resistance
Plot 1.5	BCh	Net blotch	11	89	High resistance
Plot 1.5	BCh	Brown rust	32	68	Moderate resistance
Plot 2.1	Control	Powdery mildew	55	45	Low resistance
Plot 2.1	Control	Net blotch	20	80	High resistance
Plot 2.1	Control	Brown rust	80	20	High susceptibility
Plot 2.2	MF	Powdery mildew	65	35	High susceptibility
Plot 2.2	MF	Net blotch	20	80	High resistance
Plot 2.2	MF	Brown rust	5	95	High resistance
Plot 2.3	VC	Powdery mildew	10	90	High resistance
Plot 2.3	VC	Net blotch	70	30	High susceptibility
Plot 2.3	VC	Brown rust	25	75	Moderate resistance
Plot 2.4	MF.VC	Powdery mildew	30	70	Moderate resistance
Plot 2.4	MF.VC	Net blotch	15	85	High resistance
Plot 2.4	MF.VC	Brown rust	76	24	High susceptibility
Plot 2.5	BCh	Powdery mildew	72	28	High susceptibility
Plot 2.5	BCh	Net blotch	25	75	Moderate resistance
Plot 2.5	BCh	Brown rust	10	90	High resistance

**Notes:** Control—unfertilised; MF—mineral fertiliser; VC—vermicompost; MF.VC—mineral fertiliser + vermicompost; BCh—biochar. Severity values represent field assessments of powdery mildew (*Blumeria graminis* f. sp. *hordei*), net blotch (*Pyrenophora teres* f. *maculata*), and brown rust (*Puccinia hordei*). RI was calculated as the percentage reduction in severity relative to the most susceptible treatment. Resistance categories: high resistance (RI ≥ 80%), moderate (60–79%), low (40–59%), high susceptibility (<40%).

**Table 2 plants-14-03199-t002:** ANOVA Results for the Effect of Plot Type (After Green Manure vs. After Fallow) on the Measured Parameters.

Plot Type	F-Statistic	*p*-Value	Significance (*p* < 0.05)
Plot 1 (After green manure)	0.139	0.964	No
Plot 2 (After Fallow)	0.205	0.93	No

**Notes:** The table shows F-statistics and corresponding *p*-values for comparisons between plot types. Statistical significance was considered at *p* < 0.05. No significant differences were observed.

**Table 3 plants-14-03199-t003:** Relative abundance of selected bacterial taxa across treatments in percent.

Taxonomy	*Lysobacter dokdonensis*	*Arenimicrobium luteum*	*Gemmatimonadetes bacterium* LX87		*Gemmatimonadetes bacterium* LP81	*Jahnella thaxteri*	*Agromyces ramosus*	*Steroidobacter* sp.	*Nitrospira japonica*	*Aquimonas* sp.
Control1	0.9	0.41	0.79	0.32	0.15	0.15	0.22	0.2	0.19	0.17
MF1	0.4	0.82	0.54	0.29	0.06	0.1	0.18	0.27	0.16	0.3
VC1	1.01	0.55	0.53	0.45	0.01	0.12	0.29	0.16	0.17	0.28
MF1.VC1	1.32	0.11	0.52	0.32	0.55	0.38	0.22	0.08	0.15	0.19
Bch1	0.85	0.23	0.4	0.34	0.03	0.5	0.28	0.12	0.22	0.08
Control2	0.32	0.19	0.21	0.32	0	0.26	0.17	0.33	0.32	0.11
MF2	0.56	0.26	0.42	0.37	0.02	0.3	0.23	0.28	0.29	0.17
VC2	0.54	0.17	0.45	0.4	0.04	0.22	0.29	0.2	0.18	0.08
MF2.VC2	0.79	0.14	0.66	0.56	0.05	0.24	0.35	0.13	0.21	0.17
Bch2	0.91	0.11	0.72	0.47	0	0.27	0.39	0.29	0.17	0.09

**Notes:** Values represent the proportion of each taxon within the total bacterial community based on 16S rRNA gene sequencing. Taxa are presented at genus or species level, with names written in *italic* and bold according to nomenclature standards. Treatments: Control—unfertilised; MF—mineral fertiliser; VC—vermicompost; MF.VC—mineral fertiliser + vermicompost; BCh—biochar. Numbers “1” and “2” denote plots following green manure and fallow, respectively.

**Table 4 plants-14-03199-t004:** Relative abundance (%) of selected fungal taxa across treatments.

Taxonomy	*Acrophialophora* *jodhpurensis*	*Fusarium equiseti*	*Humicola* *nigrescens*	*Chrysosporium lobatum*	*Stachybotrys chartarum*	*Humicola fuscoatra*	*Fungi* sp.	*Penicillium polonicum*
Control1	5.73	10.41	0.74	11.04	3.72	0.9	1.8	0.02
MF1	6.88	5.64	1.48	5.42	2.02	0.8	3.93	0.01
VC1	24.52	8.47	0.48	4.28	3.93	0.38	1.85	0.03
MF1.VC1	7.15	10.26	1.25	3.81	5.2	0.45	2.41	0
Bch1	7.31	14.63	0.34	3.46	3.16	0.31	3.15	0.22
Control2	0.24	4.59	11.08	5	4.85	3.97	2.32	0.15
MF2	0.23	8.39	10.1	1.92	4.77	2.92	2.14	0.13
VC2	0.19	5.62	3.02	5.3	6	4.4	2.07	0.07
MF2.VC2	0.05	8.43	13.95	4.96	6.3	2.15	1.42	2.72
Bch2	0.62	8.69	2.16	3.04	7.22	5.08	2.2	3.29

**Notes:** Values represent the proportion of each taxon within the total fungal community based on ITS amplicon sequencing. Taxa are given at genus or species level, with names written in *italic* and bold according to nomenclature standards. Treatments: Control—unfertilised; MF—mineral fertiliser; VC—vermicompost; MF.VC—mineral fertiliser + vermicompost; BCh—biochar. Numbers “1” and “2” denote plots following green manure and fallow, respectively.

**Table 5 plants-14-03199-t005:** Experimental design and fertilisation treatments applied in the two fields. Plot 1 = field established after oat–vetch green manure; Plot 2 = field established after fallow. Treatments: Control (no fertilisation), MF (mineral fertiliser), VC (vermicompost), MF.VC (combined mineral fertiliser + vermicompost), BCh (biochar).

Plot	Treatment	Description of Treatment
1	Control1	No fertilisation (reference treatment)
1	MF1	Mineral fertiliser NPK (15:15:15) at 0.1 t ha^−1^ + 50 kg N ha^−1^ (NH_4_NO_3_ at tillering)
1	VC1	Vermicompost 12 t ha^−1^
1	MF.VC1	Combined mineral fertiliser 0.05 t ha^−1^ + vermicompost 6 t ha^−1^
1	BCh1	Biochar 10 t ha^−1^
2	Control2	No fertilisation (reference treatment)
2	MF2	Mineral fertiliser NPK (15:15:15) at 0.1 t ha^−1^ + 50 kg N ha^−1^ (NH_4_NO_3_ at tillering)
2	VC2	Vermicompost 12 t ha^−1^
2	MF.VC2	Combined mineral fertiliser 0.05 t ha^−1^ + vermicompost 6 t ha^−1^
2	BCh2	Biochar 10 t ha^−1^

**Notes:** Plot 1—field established after oat–vetch green manure; Plot 2—field established after fallow. Treatments: Control—no fertilisation (reference treatment); MF—mineral fertiliser [NPK (15:15:15) at 0.1 t ha^−1^ + 50 kg N ha^−1^ (NH_4_NO_3_ at tillering)]; VC—vermicompost (12 t ha^−1^); MF.VC—combined mineral fertiliser (0.05 t ha^−1^) + vermicompost (6 t ha^−1^); BCh—biochar (10 t ha^−1^).

## Data Availability

For 16S rRNA sequencing, BioSample accessions of the analysed samples are as follows: SAMN50254222, SAMN50254223, SAMN50254224, SAMN50254225, SAMN50254226, SAMN50254227, SAMN50254228, SAMN50254229, SAMN50254230, and SAMN50254231. The data records are accessible BioProject accession number PRJNA1297989 with the submission ID: SUB15499394 using the following link: https://www.ncbi.nlm.nih.gov/bioproject/1297989, accessed on 29 July 2025. For ITS sequencing BioSample accessions of the analysed samples are as follows: SAMN50772350, SAMN50772351. SAMN50772352, SAMN50772353, SAMN50772354. SAMN50772355, SAMN50772356, SAMN50772357, SAMN50772358, and SAMN50772359. BioProject accession number PRJNA1310416 with the submission ID: SUB15563733 using the following link: https://www.ncbi.nlm.nih.gov/bioproject/PRJNA1310416, accessed on 25 August 2025.

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
