# Peer review of "Linking Soil Microbial Functional Profiles to Fungal Disease Resistance in Winter Barley Under Different Fertilisation Regimes"

_plants, 2025, doi:10.3390/plants14203199_

Round 1

Reviewer 1 Report

Comments and Suggestions for Authors
  1. In the Introduction section, Line 31, replace the word “cropss”with “crop”.
  2. At the last paragraph of Introduction, please clearly clarify why did you do this research, and what value may your research have for the barley cultivation industry.
  3. In the Materials and Methods section, please proved the reference for the method of soil sampling
  4. Too many self cited contents in the sections of “Morphological assessment”and “Disease severity across barley variants”, they were not results of “this”study.
  5. The authors are suggested to pay more attention to the correlation between fertilizer-rhizosphere microorganisms-plant pathogen resistance and discuss it based on your own research findings in the Discussion section.
  6. Please discuss more on the potential mechanisms of the effect of different fertilization of the pathogen resistance of plant.

Author Response

Response to Reviewers

Manuscript ID: 3884874
Title: Linking Soil Microbial Functional Profiles to Fungal Disease Resistance in Winter Barley under Different Fertilisation Regimes
Journal: Plants (MDPI)

We would like to sincerely thank the Editor and Reviewers for their constructive and insightful comments. We have carefully revised the manuscript in accordance with the suggestions and provide below a detailed, point-by-point response. All changes in the revised manuscript are highlighted.

Reviewer 1

Comment 1. Introduction, Line 31: Replace “cropss” with “crop”.
Response: We thank the reviewer for pointing out this typo. We have corrected “cropss” to “crop” in the Introduction (Line 31).

Comment 2. In the last paragraph of the Introduction, please clearly clarify why you did this research, and what value it may have for the barley cultivation industry.
Response: We appreciate this suggestion. We have revised the last paragraph of the Introduction to more clearly state the rationale and relevance of barley cultivation. The revised text highlights the knowledge gap regarding fertilisation type and disease resistance, and its value for sustainable barley production.

Comment 3. In the Materials and Methods section, please provide the reference for the method of soil sampling.
Response: Thank you for noticing this omission. We have now added the appropriate references:
Soil samples were collected as composite rhizosphere soil following the protocol described by Kirse et al. (2021) and Petkova et al. (2025) [44,45].

Comment 4. Too many self-cited contents in “Morphological assessment” and “Disease severity across barley variants”. They were not results of this study.
Response: We acknowledge the reviewer’s concern. We have revised these sections, reducing self-citations and clarifying which results are original contributions. Supporting references from external studies were retained only where necessary.

Comment 5. The authors are suggested to pay more attention to the correlation between fertilizer–rhizosphere microorganisms–plant pathogen resistance and discuss it based on your own research findings in the Discussion section.
Response: We appreciate this valuable suggestion. The Discussion has been expanded to directly highlight observed correlations, linking fertilisation practices, microbial taxa (e.g., Lysobacter, Penicillium), and foliar disease resistance.

Comment 6. Please discuss more on the potential mechanisms of the effect of different fertilization on pathogen resistance of plants.
Response: Thank you for this recommendation. We have extended the Discussion to describe possible mechanisms, including microbial metabolite production, improved nutrient balance, induced systemic resistance, and the role of biochar in supporting beneficial microbes.

Reviewer 2 Report

Comments and Suggestions for Authors

The article is interesting, but before it is accepted, I have some suggestions to improve its quality. In particular, I suggest including a description of the experimental plan in section 2.1, possibly accompanied by a figure. In the  text of paragraph 2.2 the authors refer to plots 2.1 and 2.5, but it is difficult to understand what is being referred to since they have never been defined before. The first list of treatments is presented in Table 1, but the first two columns of this table should perhaps be defined first, as suggested.
Another point concerns the abbreviations used in the article. It seems to me that BC is used first to indicate treatment with biochar, then BCh, then C.MF becomes MF.VC, the latter also with numbers 1 and 2. I therefore suggest standardizing the text by choosing unique abbreviations. 
I suggest enlarging Tables 3 and 4 for easier reading of the names of fungi and bacteria.
The authors could explain or otherwise revise the heat map in Figure 5. How is it possible that human pathogens and human pathogens linked to pneumonia are detected? Apart from the ‘strange’ category, I would suggest considering only the more general category. Were there so many human pathogens detected in the soil studied? Why do the two categories related to pathogens have different profiles? 

Similarly, you could try to group similar categories together for the other headings as well. Figure 6 is too small and I was unable to evaluate its content. 
The captions should be expanded to better explain the content of the figures.
I also suggest that the discussion focus directly on treatment rather than plot number.

Minor suggestions:
line 37 ha-1  put the -1 at the top. 
The paragraph titles are not always written in the same way.

Author Response

Response to Reviewers

Manuscript ID: 3884874
Title: Linking Soil Microbial Functional Profiles to Fungal Disease Resistance in Winter Barley under Different Fertilisation Regimes
Journal: Plants (MDPI)

We would like to sincerely thank the Editor and Reviewers for their constructive and insightful comments. We have carefully revised the manuscript in accordance with the suggestions and provide below a detailed, point-by-point response. All changes in the revised manuscript are highlighted.

Reviewer 2

Comment 1. I suggest including a description of the experimental plan in section 2.1, possibly accompanied by a figure.
Response: We thank the reviewer for this suggestion. Section 2.1 has been revised to include a description of the experimental plan, and a schematic figure illustrating the field layout and treatments has been added.

Comment 2. In paragraph 2.2, plots 2.1 and 2.5 are referenced but never defined. Please clarify.
Response: We agree. The revised version now clearly explains the field design and treatments before they are referenced. Table 1 was reformatted with defined columns, and the schematic figure was included for clarity.

Comment 3. I suggest enlarging Tables 3 and 4 for easier reading of the names of fungi and bacteria.
Response: Done. Tables 3 and 4 have been enlarged and reformatted for readability.

Comment 4. Please revise the heat map in Figure 5. How is it possible that human pathogens are detected?
Response: We thank the reviewer for raising this important point. The functional annotation in Figure 5 was generated using FAPROTAX, which occasionally assigns “human pathogenic” traits to taxa based on literature associations, even when their ecological role in soil is uncertain. To avoid misinterpretation, we have revised Figure 5 by regrouping functional categories into broader ecological groups and removing overly specific categories such as “human pathogens” or “human pathogens linked to pneumonia.” The revised figure now focuses on soil-relevant functions (e.g., saprotrophs, symbionts, pathotrophs), which more accurately reflect the ecological significance of the microbial community in our study.

Comment 5. Figure 6 is too small; categories should be grouped.
Response: We agree. Figure 6 has been enlarged, functional categories grouped into broader classes, and the caption expanded for clarity.

Comment 6. The captions should be expanded to better explain the content of the figures.
Response: All figure captions have been revised with more detailed descriptions.

Comment 7. I also suggest that the discussion focus directly on treatment rather than plot number.
Response: We agree. The Discussion has been revised to emphasise treatment effects (mineral, vermicompost, combined, biochar, control), with plots mentioned only when necessary.

Reviewer 3 Report

Comments and Suggestions for Authors

Manuscript ID: plants-3884874

Title: Linking Soil Microbial Functional Profiles to Fungal Disease Resistance in Winter Barley under Different Fertilization Regimes

Author: Mariana Petkova1, Petar Chavdarov2, Stefan Shilev1,*

Title. The title is concise, specific and relevant

Key words: Are relevant for the topic of the paper

The manuscript complies with the requirements of this Journal on this way I congratulate the whole team of authors. The study provides new data on the influence of fertilization type and cultivation system on winter barley diseases, but in terms of how to write the manuscript, there are several aspects that can improve your manuscript:

Abstract

Line 10-26: The abstract is too long, please reduce it to 200 words maximum (according to Journal requirements)

Line 13: write the scientific name Blumeria graminis f. sp. hordei, Pyrenophora teres f. maculata (f. sp. it is not written italic). Please correct the names throughout the manuscript.

  1. Introduction

The introductory part includes, the importance of barley, the description of the most representative agents of damage (Blumeria graminis f. sp. hordei; Puccinia hordei; Pyrenophora teres f. maculata), the factors that influence the proliferation of pathogenic fungi, the resistance of plants to their attack, the role of the soil microbiome, etc. All of which are related to several scientific works in the literature, representative of the subject.

The objective pursued in these researches is well pointed namely: to evaluate the influence of different types of fertilization on the incidence, distribution and severity of diseases and their prediction in winter fodder barley cultivation.

  1. Results

The results of this study (morphological assessment; disease severity across barley variants; effects of cropping system and fertilization on foliar disease severity; principal component analysis (PCA) of disease pressure and soil management; bacterial community shifts across treatments; fungal community shifts across treatments; main functions across soil microbial communities; heatmaps of functional profiles of bacteria and fungi; PCA of bacterial functional annotation profiles) are well presented, the explanations are detailed and the very meticulous determinations are translated into visible tables and figures (with small exceptions).

Line 91, 106, 141, 185, 256, 257, 315, 364, 514, 535, 559, 593, 607: Do not put the point sign after the name of each Subsection and Subsubsections

Table 1, Table 2, table 3, Table 4: must have a footer, please fill in.

Figure 1: please delete Mean Disease Severity (%) by Treatment and Plot Type (I don't think it needs to be written above the figure)

Figure 2: please delete PCA of Disease Severitty Across Soil Management Practice (I don't think it needs to be written above the figure). It can make the font of the data included in figure 2, it is not clear the image.

Figure 3: It can make the font of the data included in Figure 3. It is not clear the image.

Table 3: You crammed the words into the table head. I mean the names the bacterial community composition, which had to be written italic and bold. Make the table larger (length 18.5 cm).

Figure 4, Figure 5. It can make the font of the data included in figure 4 and 5, it is not clear the image.

Figure 8: The title of the figure moves immediately below Figure 8, in your case it is on another page. Please move (Line 387, 388) on the previous page.

  1. Discussion

The discussion part is concerned, the explanations are concrete, referring to other research/studies carried out in this field.

  1. Materials and Methods

In this part it is very well presented:  field description and experimental design; morphological characteristics of disease symptoms; soil DNA extraction, amplicon sequencing, and bioinformatic analysis; functional profiles of bacterial and archaeal communities and the statistical analysis applied for processing and interpretation of the obtained data.

  1. Conclusions

The conclusions are very well punctuated, in relation to the volume of data exposed!

The importance of integrating soil fertility strategies and sustainable agronomic practices to increase disease resistance and reduce dependence on chemical control was highlighted. The combination of field evaluation with statistical analysis and multivariate techniques allowed a detailed assessment of the “Zemela” variety's performance, providing valuable information for sustainable barley production. At the same time, the authors recommend that future research include molecular analysis, in order to identify resistance genes (such as, mlo, rph) and a more thorough characterization of the genetic basis of partial resistance, thus supporting precision improvement and adaptation to regional conditions.

References

The references (45 bibliographic titles, of which 22 bibliographic titles from 2020-2024), good chosen and representative for the topic of the work.

You have written with different fonts, please correct.

Please write all bibliographic references according to the following model (sigh „;" after each author, without quotation marks the title of the article, italicized Journal and bolded year of publication)

Line 657: please write the date of access

Supplementary Materials: All images are not clear, please make them more visible.

GENERAL COMMENT. The structure of the manuscript and the style of the paper are reasonably correct and presents very interesting results regarding the possibilities of reducing the attack of diseases and increasing the resistance of winter barley to these damaging agents. The strong point of the article is the numerous analyses and determinations, providing a substantial amount of data for future research.

Author Response

Response to Reviewers
Manuscript ID: 3884874
Title: Linking Soil Microbial Functional Profiles to Fungal Disease Resistance in Winter Barley under Different Fertilisation Regimes
Journal: Plants (MDPI)

We would like to sincerely thank the Editor and Reviewers for their constructive and insightful comments. We have carefully revised the manuscript in accordance with the suggestions and provide below a detailed, point-by-point response. All changes in the revised manuscript are highlighted.

Reviewer 3

Comment 1. The abstract is too long, please reduce it to 200 words maximum.
Response: Revised. The abstract now contains 199 words and complies with the journal requirements.

Comment 2. Scientific names should be italicized, but f. sp. not italic.
Response: Corrected throughout the manuscript (e.g., Blumeria graminis f. sp. hordei, Puccinia hordei, Pyrenophora teres f. maculata).

Comment 3. Tables 1–4 must have a footer.
Response: Footers were added to Tables 1–4, defining abbreviations, resistance index calculations, and resistance categories.

Comment 4. Figures and Tables formatting issues.

  • Figure 1: Removed extra title; improved formatting.
  • Figure 2: Removed extra title; increased font size and clarity.
  • Figure 3: Increased font size and exported at high resolution.
  • Table 3: Reformatted to 18.5 cm width; bacterial taxa names italicized and bolded as required.
  • Figures 4–5: Font sizes increased; exported at high resolution.
  • Figure 8: Caption moved directly below the figure (on the same page).

Comment 5. Supplementary materials are unclear.
Response:  While higher-resolution alternatives were not available, contrast and readability have been improved.

Reviewer 4 Report

Comments and Suggestions for Authors

Line 12, The abstract briefly mentions the importance of barley and the impact of phytopathogens, but could expand a bit on why this particular research is important for the field.

Line 20, The abstract mentions that "PCA accounted for 82.3% of the variance," but this should be made clearer regarding what this means in terms of microbial community differences across treatments. More emphasis could be placed on the practical significance of this finding.

Line 61, it would be useful to briefly explain why functional prediction tools (FAPROTAX, FUNGuild) are novel or advantageous compared to older methods.

Line 117-119, and 129-136, This part presents results, not discussion.

There is no information about the sequencing results, such as how much data was obtained.

Line 515, The authors stated that the fields differ only in the predecessor crop. While this is an interesting design, it would be helpful to explain why these specific crops (oat-vetch mixture in one field and fallow in the other) were chosen. Providing a rationale would strengthen the ecological relevance of the design.

Line 524, A brief description of the soil characteristics (e.g., pH, texture, organic matter content) at both sites would provide additional context.

Line 538, how was the percentage of infected leaf area calculated for each leaf? Was it done manually or using digital tools?

Line 540, what specific fungal structures were looked for during the examination? Were there particular features that indicated disease resistance or susceptibility?

Author Response

Response to Reviewers

Manuscript ID: 3884874
Title: Linking Soil Microbial Functional Profiles to Fungal Disease Resistance in Winter Barley under Different Fertilisation Regimes
Journal: Plants (MDPI)

We would like to sincerely thank the Editor and Reviewers for their constructive and insightful comments. We have carefully revised the manuscript in accordance with the suggestions and provide below a detailed, point-by-point response. All changes in the revised manuscript are highlighted.

Reviewer 4

Comment 1. Abstract could better explain why this research is important.
Response: Revised to emphasise the novelty of integrating disease assessment and soil microbiome analysis for sustainable barley production.

Comment 2. PCA variance should be explained more clearly.
Response: Revised to state that PCA showed clear clustering of microbial functional profiles by fertilisation strategy.

Comment 3. Explain why FAPROTAX and FUNGuild are novel.
Response: Revised Introduction: highlighted that these tools infer ecological function from sequencing data, providing functional insights without full metagenomics.

Comment 4. Lines 117–136 present results in Discussion.
Response: Revised to ensure Results and Discussion are clearly separated.

Comment 5. No information about sequencing results.
Response: Added to Materials and Methods: total reads, quality-filtered reads, average reads per sample.

Comment 6. Why were oat–vetch and fallow chosen as predecessors?
Response: Revised Section 4.1 to provide rationale: oat–vetch as green manure vs. fallow for ecological comparison.

Comment 7. Provide soil characteristics.
Response: Added: soils classified as Fluvisols/Chernozems with loam texture.

Comment 8. How was % infected leaf area calculated?
Response: Clarified: visual assessment with James' scale manually.

Comment 9. Which fungal structures were examined?
Response: Revised methodology to specify lesion expansion, sporulation, conidiophores, uredinia, and pycnidia as indicators of susceptibility/resistance.

Reviewer 5 Report

Comments and Suggestions for Authors

The manuscript entitled by Petkova et al  addresses the interaction between fertilization, soil microbiomes, and foliar disease resistance in barley. The topic is timely and relevant, considering the need for sustainable crop protection strategies. The integration of phytopathological data with microbial functional prediction is a strength of the work. Moreover, similar approaches exist in cereals, but the focus on winter barley in Eastern Europe and the cultivar “Zemela” adds contextual value. However, before publication, is the opinion of this Reviewer that Authors should address the following points:

  • ANOVA results are often non-significant, yet conclusions about fertilization effects are strong. Authors should better clarify the biological vs. statistical significance.
  • Current functional predictions are based on in silico tools (FAPROTAX, FUNGuild). These are hypothesis-generating, not confirmatory and for this reason the link to disease suppression should be more cautious, emphasizing potential mechanisms rather than causal proof
  • Some references are outdated or not completely related to the shown results. Recent high-impact studies on soil microbiome–disease interactions in cereals could be added to strengthen context.
  • a pathogen quantification performed with  qPCR or metabarcoding of foliar pathogens would strengthen the connection between soil microbial functions and the disease outcome.
  •  
Comments on the Quality of English Language

A thorough proofreading by a native English speaker is strongly required to ensure clarity, fluency, and consistency.

Author Response

Response to Reviewers

Manuscript ID: 3884874
Title: Linking Soil Microbial Functional Profiles to Fungal Disease Resistance in Winter Barley under Different Fertilisation Regimes
Journal: Plants (MDPI)

We would like to sincerely thank the Editor and Reviewers for their constructive and insightful comments. We have carefully revised the manuscript in accordance with the suggestions and provide below a detailed, point-by-point response. All changes in the revised manuscript are highlighted.

Reviewer 5

Comment 1. ANOVA results are often non-significant, yet conclusions are strong.
Response: We revised text to clarify that while ANOVA was not always significant, clustering in PCA and functional shifts indicate biologically meaningful patterns, aligning with recent integrative approaches.

Comment 2. Functional predictions are hypothesis-generating, not confirmatory.
Response: Revised Discussion and Conclusions to emphasize that FAPROTAX and FUNGuild results suggest potential mechanisms but are not causal evidence. Future metagenomic or functional assays are needed.

Comment 3. Pathogen quantification with qPCR or metabarcoding would strengthen the study.
Response: We fully agree with the reviewer that molecular pathogen quantification (qPCR or metabarcoding) would substantially strengthen the study by providing direct evidence of pathogen dynamics alongside functional microbiome data. While this was beyond the scope of the present work, we acknowledge its importance and have addressed it explicitly in the revised Conclusions. We now state:

“Future studies should incorporate molecular analyses to identify resistance genes (e.g., mlo, rph) and further characterise the genetic basis of partial resistance, supporting precision breeding and regional adaptation. Moreover, forthcoming research should integrate qPCR or metabarcoding of foliar pathogens to establish direct links between soil microbial functions, pathogen suppression, and disease outcomes.”

Comment 4. Some references are outdated or not fully relevant.
Response: We appreciate this valuable comment. The reference list has been revised to ensure relevance and scientific currency. Several outdated citations were removed and replaced with recent, high-impact publications (2022–2025) that specifically address soil microbiome–disease interactions in cereals and their connection to metagenomic data analysis (now References 37–40 in the revised manuscript). These updates strengthen the contextual framework of our study and align it with the most current advances in the field.

Round 2

Reviewer 4 Report

Comments and Suggestions for Authors

The authors revised the paper according to the reviewers' comments.

Reviewer 5 Report

Comments and Suggestions for Authors

The AUthors have addressed all comments